# Geographic proximity to primary care providers as a risk-assessment criterion for quality performance measures

Nathaniel Bell[1‡]*, Ana Lòpez-De Fede[2‡], Bo Cai[3⊙], John Brooks[3⊙]

**1** College of Nursing, University of South Carolina, Columbia, South Carolina, United States of America, **2** Institute for Families in Society, University of South Carolina, Columbia, South Carolina, United States of America, **3** Arnold School of Public Health, University of South Carolina, Columbia, South Carolina, United States of America

⊙ These authors contributed equally to this work.
‡ NB and AL-DF are joint senior authors on this work and also contributed equally to this work
* nateb@mailbox.sc.edu

## Abstract

### Importance

Previous studies have found a mixed association between Patient-Centered Medical Home (PCMH) designation and improvements in primary care quality indicators, including avoidable pediatric emergency department (ED) encounters. Whether these associations persist after accounting for the geographic locations of providers relative to where patients reside is unknown.

### Objective

To examine the association between geographic proximity to primary care providers versus hospitals and risk of avoidable and potentially avoidable ED visits among children with pre-existing diagnosis of attention-deficit/hyperactivity disorder or asthma.

### Methods

Retrospective cohort study of a panel of pediatric Medicaid claims data from the South Carolina from 2016–2018 for 2,959 beneficiaries having a pre-existing diagnosis of attention-deficit/hyperactivity disorder (ADD, ages 6–12) and 6,390 beneficiaries with asthma (MMA, ages 5–18), as defined using Healthcare Effectiveness Data and Information Set (HEDIS) performance measures. We calculated differences in avoidable and potentially avoidable ED visits by the beneficiary's PCMH attribution type and in relation to differences in proximity to their primary care providers versus hospitals. Outcomes were defined using the New York University Emergency Department Algorithm (NYU-EDA). Differences in ED visit risk were assessed using generalized estimation equations and compared using marginal effects models.

### Results

The 2.4 percentage point reduction in risk of avoidable ED visits among children in the ADD cohort who attended a PCMH versus those who did not increased to 3.9 to 7.2 percentage

**Data Availability Statement:** Data cannot be shared publicly because of HIPAA Privacy Rules, but the strategies for linkages analysis analyses can be provided through contacting the authors.

Data are available from the University of South Carolina IRB (contact Lisa Johnson at lisaj@mailbox.sc.edu) for researchers who meet the criteria for access to confidential data.

**Funding:** This research is funded by the Agency for Healthcare Research and Quality (AHRQ R03HS026263-01A1). All authors were named as applicants for this award. The funders had no role in the study design, data collection, or analysis nor did the funders participate in writing this manuscript or have any decision to publish this study.

**Competing interests:** The authors have declared that no competing interests exist.

**Abbreviations:** ADD HEDIS, measure for Follow-Up Care for Children Prescribed ADHD Medication; MMA HEDIS, measure Medication Management for People with Asthma; FFS A, Medicaid beneficiary who is enrolled under a fee for service payment plan; NCQA, National Committee for Quality Assurance; GIS, geographic information systems/science; HEDIS, Healthcare Effectiveness Data and Information Set; PCMH, Patient-Centered Medical Homes; Non-PCMH, Medicaid beneficiaries who were un-enrolled and did not attend a designated medical home.

points as relative proximity to primary care providers versus hospitals improved (p < 0.01). Children in the ADD and MMA cohorts that were enrolled in a medical home, but did not attend one for primary care services exhibited a 5.4 and 3.0 percentage point increase in avoidable ED visit compared to children who were unenrolled and did not attend medical homes (p < 0.05), but these differences were only observed when geographic proximity to hospitals was more convenient than primary care providers. Mixed findings were observed for potentially avoidable visits.

## Conclusions

In several health care performance evaluations, patient-centered medical homes have not been found to reduce differences in hospital utilization for conditions that are treatable in primary care settings among children with chronic illnesses. Analytical approaches that also consider geographic proximity to health care services can identify performance benefits of medical homes. Expanding risk-adjustment models to also include geographic data would benefit ongoing quality improvement initiatives.

## Introduction

Ample evidence shows that health outcomes across the United States (US) differ by race, ethnicity, and socioeconomic status [1–5]. For instance, emergency department (ED) utilization is higher among the homeless [6], diabetes prevalence is often attributed to greater food insecurity [7], and social isolation contributes to both stroke and heart disease [8], to name but a few of its determinants and consequences. It is also possible to make a reasonably good prediction as to how long it will take people to access care or the type of treatment they will receive based on knowing where they live [9–11]. For example, recent studies have shown that Medicaid beneficiaries who live closer to EDs have a higher probability of using these services for non-emergent reasons than other patient groups [12].

Inappropriate or non-emergent ED utilizations raise costs, strain resources, as well as increase wait times for care [13,14]. They are also fundamentally linked to social determinants of health, such as poor primary care access or low health literacy [15], as well as to care management on the part of patients and primary care providers [16]. Yet, despite national recommendations on the general management of chronic illnesses and the substantial cost and negative implications of non-emergent ED visits, pre-COVID 19 pandemic studies found that the frequency of avoidable visits were increasing, particularly among children [17]. For example, it is estimated that nearly 75% of the 3 million ED visits for asthma-related care and nearly 33% of the 1.6 million ED visits for mental health-related conditions can be treated in primary care settings [18–21]. Medicaid is often the most frequent payer of ED visits nationwide [22].

Over the past decade, patient-centered medical homes (PCMH) have shown potential to limit avoidable ED utilizations and improve preventative and chronic disease management [23,24]. Akin to other patient-based practice structures, such as Canada's Family Health Teams [25], medical homes are largely based on a physician-led continuity of care model that include nurses, nutritionists, pharmacists, case managers, and social workers with an emphasis on providing patient-and culturally-centered care [26]. In the broadest sense, they represent a transformative whole-person approach to primary care designed to improve physical health, behavioral health, access to community-based social services, as well as better management of chronic

illnesses [27–30]. Aspects of medical home design, such as its use of care coordinators to help with medication reconciliation are particularly relevant for children with chronic conditions who have a diverse set of medical needs [31]. For instance, case managers may work directly with children and their guardians to coordinate care, help families navigate the health care system, as well as coordinate appointments with specialists to improve disease control [32].

Although there is growing consensus of the benefit of PCMH-modeled care on health outcomes overall, including lowered ED utilization, its effectiveness in reducing pediatric ED utilizations have been mixed. Among the general pediatric population, evaluations have shown a reduction in ED use as well as improvements in preventative care delivery [33–35], but typically show little effect on usage rates among children with chronic illnesses, even when evaluated using patient-reported quality metrics (e.g., wait times to care, satisfaction with providers) [36–38]. Mixed findings are thought to stem from the amplified effect that care fragmentation and poor communication across resource systems have on children with greater medical needs [39]. It is also possible that a caregiver's perception that their child is not receiving sufficient care may be a stronger predictor of ED use regardless of the quality of primary services the child receives. However, missing from previous evaluations is whether the interaction between where patients live and where their health care services are located modifies these associations. Although some PCMH evaluation studies have attempted to control for geography (e.g., rural versus urban clinics) [40,41], this approach ignores the spatial connectivity between patients and providers and cannot answer questions as to whether greater travel distances to care determines why the effects of some medical home innovations are often muted.

In this study, we assessed whether geographic proximity to health care providers modified the association between medical home attendance and risk of avoidable and potentially avoidable ED visits among pediatric Medicaid recipients in South Carolina (SC) having a pre-existing diagnosis of asthma or attention-deficit/hyperactivity disorder, each of which are chronic illnesses that often result in ED visits for disease-specific care that is often treatable in primary care settings [42,43]. Each condition was defined using the National Committee for Quality Assurance's (NCQA) Healthcare Effectiveness Data and Information Set (HEDIS®) process measures, thus reflecting a measurement standard commonly used for risk adjustment and performance tracking by state Medicaid agencies. Although many initial evaluations of the PCMH transformation on outcomes among Medicaid participants have been positive, pointing to the benefit of better compliance measures, case management, and electronic health records with earlier and better access to care [44–47], some studies have failed to show consistent improvements quality [48,49]. Fig 1 provides an example of why spatial interactions in these and other PCMH evaluations might be warranted. It shows that SC counties that have witnessed the greatest penetration of medical homes are simultaneously among the least dense with respect to the % of the population enrolled in Medicaid (areas in bright green). What is also particularly telling is the 'I-95 corridor', often referred to as the 'corridor of shame' [50], which stretches from Jasper county in the south to Dillon county in the northeast, is among the areas with the poorest access (areas in bright pink). Eight of the 17 counties that form part of the corridor are on the wrong end of the PCMH availability-to-need spectrum, many of which are among the poorest counties in the country.

Given what is known about proximity to care as a determinant of service utilization and outcomes, statistical models that include interactions with spatial data representative of where patients reside relative to where they receive primary and emergency care services may help reveal unmeasured benefits or barriers attributed to medical home-modeled care. Assessing the representation of one type of service accessibility measure (i.e., travel distances) is important for ongoing policy decisions for state Medicaid programs, which almost universally provide support, technical assistance, and resources to clinicians, practices, and families to ensure

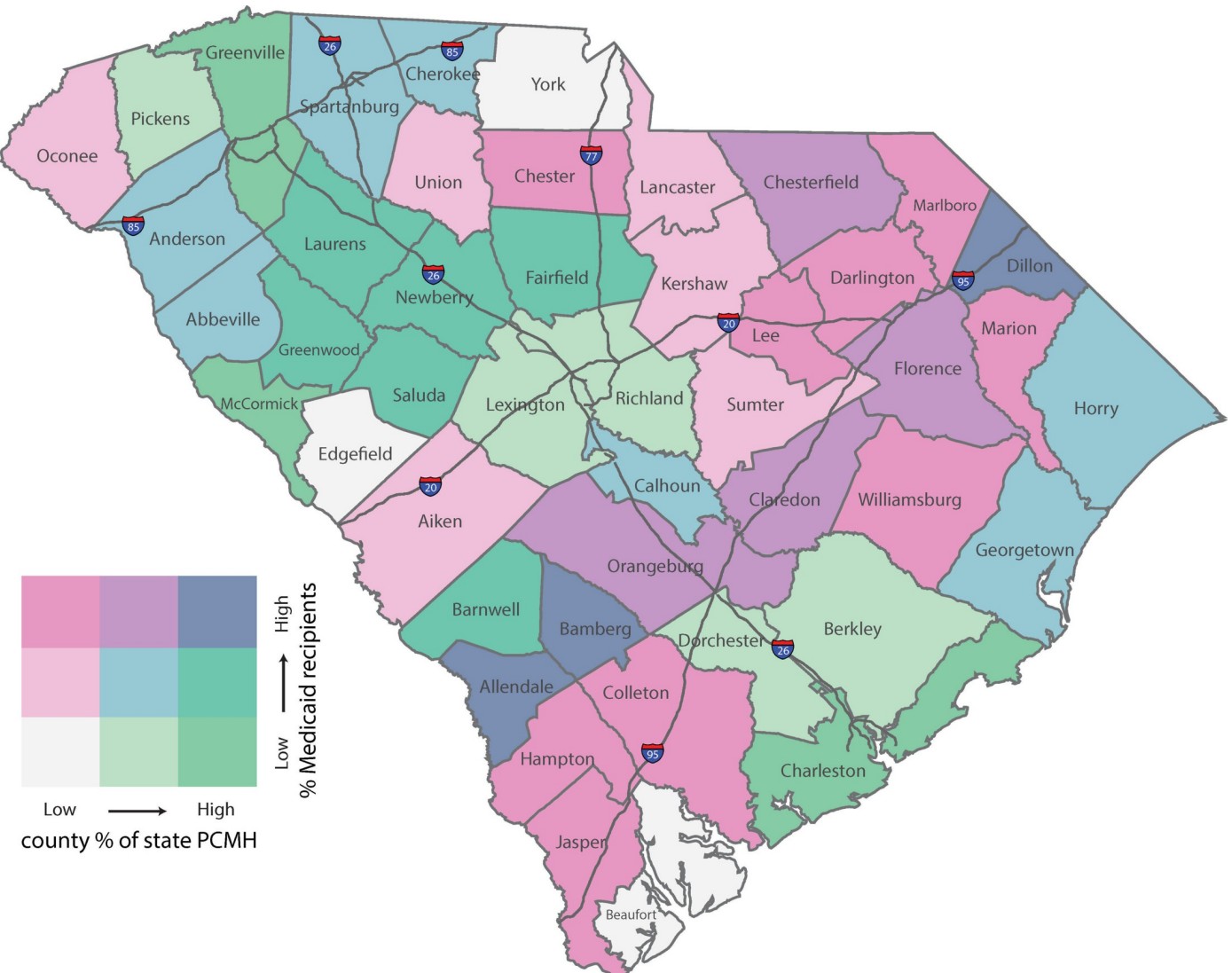

**Fig 1. Bivariate choropleth map of county proportion of medical homes relative to all primary care providers and the proportion of county population enrolled in Medicaid.** Figure created by the authors using the University of South Carolina's site license for ArcMap. Data to recreate the map is publicly available through the American Community Survey, the NPI Certification files distributed by the Centers for Medicare and Medicaid Services, and the NCQA data feed file (NCQA data available through subscription).

its beneficiaries gain access to designated medial homes [51]. This emphasis is also timely given the ongoing transformations of US risk-adjustment and pay-for-performance models to include social factors on top of the clinical risk factors already in place [52–54]. The NCQA is one of the most recent organizations to propose health plan accountability for social risk factors [55]. The NCQA is also one of the largest accrediting bodies that certifies primary care clinics as designated medical homes.

## Materials and methods

### Patient population

Approximately one out of every five of the 5.1 million persons residing in SC is a Medicaid beneficiary, of which 62% are 18 years of age or younger [56]. In 2013, the SC Department of

Health and Human Services' (SCDHHS) Medicaid agency began working with its managed care programs to enroll eligible beneficiaries into designated PCMHs. The intention has been to improve access to primary care, particularly care that is considered to be of higher quality. This study used a 3-year retrospective cohort design of pre-COVID 19 pandemic data, beginning with a baseline (reference) year of 2016, which is consistent with other Medicaid program effectiveness evaluations [57]. Medicaid claims were eligible for inclusion if the beneficiary was continuously enrolled for the entire 2016 reporting year, but the patient was not required to be enrolled for the entire study period. Continuous enrollment in South Carolina allows patients to churn on/off Medicaid for no more than 45 days per year.

We used the NCQA's Healthcare Effectiveness Data and Information Set (HEDIS®) process measures to define a similar patient cohort and as a grouping mechanism to differentiate differences in outcomes achieved from designated and non-designated clinics. HEDIS® measures comprise the core evaluation metrics for pediatric and adult Medicaid programs and relate to numerous public health issues, including cancer, heart disease, mental health, asthma, and diabetes, among others. HEDIS® metrics are used to measure specific aspects of primary care quality, including the effectiveness (e.g., colorectal cancer screening) and experiences of care (e.g., CAHPS surveys), access/availability of care (e.g., access to ambulatory health services), risk adjusted utilizations (e.g., well-child visits), as well as relative resource use (e.g., resource use for people with COPD). In 2016, 37 state Medicaid programs collected or required its Managed Care plans to report on over 60 HEDIS® measures as part of annual external quality reviews [58]. The use of HEDIS® metrics as a grouping mechanism for PCMH evaluations is also common practice when attempting to differentiate outcomes by provider settings [59,60].

Our patient sample included all Medicaid beneficiaries that met the inclusion criteria for the HEDIS® measure of follow-up care for attention-deficit/hyperactivity disorder (ADD) or medication management for people with asthma (MMA). The ADD measure is specific to recipients between 6 and 12 years of age who were newly prescribed ADHD medication. The MMA measure captures child recipients between the ages of 5 and 18. For this analysis, we collapsed the MMA age ranges into a single group. These measures were chosen because they represent conditions that result in frequent presentation in hospital emergency departments for disease-specific care that is often treatable in primary care settings [42,43]. Both measures were also chosen because they have a strong association with social vulnerability and residing in economically deprived areas and may be part of expanded risk factor adjustment metrics [55]. The study protocol (#Pro00093322) was reviewed and approved by the University's Institutional Review Board (IRB) in accordance with 45 CFR 46.104(d)(4).

## PCMH attribution

SC Medicaid recipients select their primary care provider from five similarly designed Managed Care Organizations (MCO) or through fee-for-service (FFS) [61]. Over 80% of all Medicaid beneficiaries in SC are enrolled in an MCO, the majority of which include clinics that are NCQA-designated medical homes. This is in line national recommendations from the American Academy of Pediatrics (AAP), the National Association of Pediatric Nurse Practitioners (NAPNAP), and other organizations that all children have access to comprehensive, accessible, coordinated, and culturally appropriate care [62,63]. At the time of this study, there were no health (e.g., the number of chronic conditions) or plan-related (enrolled in MCO or FSS) restrictions on SC Medicaid beneficiaries for selecting a primary care provider whose practice was or was not a designated/accredited medical home.

It was not possible to identify from the claims data how each beneficiary initially elected to enroll with their primary care provider. However, it was possible to identify from the

claims data whether the recipient's health plan assigned them to a medical home as well as determine the designation status of the their primary care provider using their National Provider Identifier (NPI) and the NCAQ provider file. As such, we modeled two different depictions of PCMH attribution on pediatric ED visits: one based solely on attendance (i.e., attending a PCMH or not attending a PCMH) and one based on a 2x2 matrix of enrollment and attendance combinations (e.g., enrolled and attended a PCMH, enrolled and never attended a PCMH, etc.). We assessed both attribution types as the first method provides an account of care quality when based solely on patient attendance patterns that are typically available in health care registries, whereas the latter provides a finer depiction of a medical home effect in relation to geography. Beneficiaries that did not attend a PCMH or who were un-enrolled in a PCMH and never attended one for their primary care were used as the reference group.

### Emergency department visits

We used the New York University Emergency Department Algorithm (NYU-EDA) to define the level of acuity for all ED visits [64]. The NYU-EDA is a widely used measure of avoidable ED utilization [65–70]. The NYU-EDA classifies each ED visit probabilistically on a percentage basis into one of four categories: "non-emergent," "emergent, but primary care treatable," "emergent, but preventable/avoidable," and "emergent, not preventable/avoidable". All visits are classified based on the ICD diagnosis codes from the patient's chart. The algorithm excludes uncommon diagnoses and treats mental health and substance abuse diagnoses separately from other causes. As it is possible to have probability scores assigned across all 4 categories, we used modifications to the event diagnosis groupings validated by Ballard et al (2010) to assign "non-emergent" (i.e., avoidable) and "emergent, but primary care treatable" (i.e., potentially avoidable) events into two distinct group, so long as the visit probability was greater than 0.50 for either event [71]. We excluded all visits that were the result of injury, or alcohol or drug problems since these conditions usually require ED services regardless of severity.

### Covariates

We were able to identify a limited set of covariate information from the claims data, including age, sex, race/ethnicity, co-morbidity classification, number of primary care visits, whether the ED visit occurred on a weekend, as well as whether the claim was attributed as fee-for-service (FFS). At the time of this study, the race/ethnicity field was not a mandatory data collection point by the South Carolina Medicaid agency. Due to small numbers and the large number of unknown/missing responses, we included only beneficiaries who reported their race/ethnicity as non-White Hispanic, Black, or Hispanic. Co-morbidities were defined using 3M$^{TM}$ clinical risk group classification codes (CRG) [72]. CRGs are a population classification system derived from inpatient and ambulatory diagnosis and procedure codes, pharmaceutical data, as well as patient functional health status. CRG codes categorize patients into one of nine severity-adjusted groups, increasing in scale from heathy/non-users to catastrophic condition status. Through data linkages, we were able to obtain additional information pertaining to the patient's neighborhood dwelling area type using modifications of the Rural Urban Commuting Area (RUCA) codes, the median household income of their residential census tract, as well as the proportion of medical homes relative to all primary care providers within their county of residence. RUCA codes were fixed using 2010 estimates whereas income and provider ratios were calculated yearly using American Community Survey and NCQA reports.

## Geographic assessment

To calculate geographic distances between beneficiaries and providers we built a composite geocoding algorithm that allowed for situs, linear, and area referencing of Medicaid claims data using a previously validated approach [73]. Prior to geocoding, we standardized each address file to US Postal Service mailing format to increase the likelihood of matching the provider address information to the street centerline file. Standardization was done using ZP4 address correction software. Next, we linked address information (e.g., street name, street suffix, ZIP code) from the claims database to street centerline data using ESRI's Street Map Premium Address File. Average travel distance estimates (in miles) for all visits were then constructed for all unique pairings between the beneficiary's residential address recorded in the claims database to the location where their primary care and ED visit occurred. The distance metric used in the regression model was the difference in distances between a beneficiary and their primary care provider versus the ED where the visit occurred. Negative values represented more convenient access to primary care providers, whereas positive values represented better proximity to hospitals. For ease of interpretation, relative differences in access were grouped using +/- 15 mile increments, with values of 0 representing no difference in distance between either location relative to the beneficiary's residential address.

## Statistical analysis

In the unadjusted analysis, differences between means of continuous variables across PCMH attribution classifications were compared using ANOVA. Differences in proportions of categorical variables were examined using chi square tests. In the adjusted models, we measured the association between PCMH attribution and ED visits in two ways. First, we constructed a-spatial regression models to assess adjusted differences in event probabilities based on a patient's PCMH attribution group type, with one model comparing ED visits based solely on PCMH attendance (e.g., attended a PCMH versus non-PCMH attendees) and another comparing visits based on the enrollment and attendance matrix (e.g., enrolled and attended a PCMH, enrolled and never attended a PCMH, etc.). Second, we expanded each model to account for the interaction with travel distance to providers. The spatial interaction models included all previously adjusted covariates.

We used a logistic regression analysis and marginal effects classifications to measure and compare the association between PCMH attribution type and the probability of an avoidable or potentially avoidable ED visit. Since beneficiaries may have had multiple ED visits during the study period as well as may not have been enrolled for the entire 3-year study period, we used a logistic model within a GEE to examine the probability of events while explicitly modeling the within-subject correlation. We initially selected a model using an independent correlation structure due to the small clusters and imbalanced design as well as the lack of complete confounder adjustment from the claims database. We ran quasi likelihood criterion tests to confirm this correlation structure was the best for model fit. All covariates that were statistically significant ($p < 0.10$) in bivariate analysis for each cohort were included in the regression models as a potential influence on ED visit risk. A full list of the covariates and regression coefficients from the models are provided in the online supplementary appendix. All statistical analyses were completed in STATA for Windows.

## Results

The final ADD patient sample consisted of 4,222 person-years from 2,959 unique non-Hispanic White, Black, and Hispanic Medicaid beneficiaries. From a total of 6,059 ED visits among the ADD cohort, 1,355 (22.3%) were classified as being non-emergent (i.e., avoidable)

and 1,026 (16.9%) were classified as being emergent but primary care treatable (i.e., potentially avoidable), for an avoidable and potentially avoidable ED visit rate of 32.1 and 24.3 visits per 100 person-years, respectively. The final MMA patient sample consisted of 10,998 person-years from 6,390 unique non-Hispanic White, Black, and Hispanic Medicaid beneficiaries. There was a total of 20,462 ED visits, of which 4,205 (20.6%) and 3,515 (17.2%) were classified as avoidable or potentially avoidable, for a person-year rate of 38.2 and 31.9 avoidable and potentially avoidable ED visits per 100 person-years, respectively.

Tables 1 and 2 show that across all patients and PCMH attribution groups, the cohorts were dissimilar with respect to age, race/ethnicity, number of primary care provider visits, RUCA dwelling type, and county provider ratios (p < 0.05). The travel distances to primary care providers and hospitals were dissimilar among the MMA cohort for both PCMH attribution types, with medical home patients residing closer to all services irrespective of how enrollment and attendance groupings were defined (p < 0.01). There were no statistically significant differences in average travel differences to any service provider for the ADD cohort in the unadjusted comparisons.

A-spatial adjusted predicted probabilities for ED visit type by clinic attribution grouping for each cohort are shown in Table 3. Among the ADD cohort, the adjusted predicted probability of experiencing an avoidable or potentially avoidable visit by clinic attribution grouping ranged from 0.205 to 0.282 and from 0.162 to 0.174, respectively. The 2.4 percentage point reduction in risk of avoidable ED visits among children in the ADD cohort who attended a PCMH versus those who did not was statistically significant (p < 0.01). The 1.1 percentage point increase in risk of a potentially avoidable ED visit among children in the ADD cohort who attended a PCMH was not statistically significant. When PCMH attribution was based on enrollment and attendance, children in the ADD cohort who were enrolled in a PCMH but never attended one for their primary care exhibited a 5.4 percentage point increase in an avoidable ED visit (p < 0.05) compared to children who were un-enrolled in a medical home and never attended one for primary care (reference group). The 2.4 percentage point decrease in risk for avoidable visits among children that were enrolled and attended a medical home was not statistically significant from the reference group. Among the MMA cohort, the adjusted probability for avoidable or potentially avoidable visits ranged from 0.196 to 0.226 and from 0.167 to 0.173 across all attribution groups, respectively. Children in the MMA cohort who were enrolled in a PCMH and never attended one for primary care exhibited a 3.0 percentage point increase in risk of an avoidable ED visit compared to children in the reference group (p < 0.05). No other comparisons were statistically significant. Regression coefficients for all models are provided in the supplementary online appendix.

Table 4 shows the results of models that included the spatial interactions. After accounting for relative differences in proximity to providers, the 2.4 percentage point decrease in risk of avoidable ED visits among children in the ADD cohort shown in Table 3 increased to 3.9 to 7.2 percentage points as the relative proximity to primary care providers over hospitals improved (p < 0.05). The differences in risk remained 3.1 percentage points lower so long as the distance to the medical home was not 15 miles further to access than the hospital (p < 0.01). In contrast to the a-spatial model that found no difference in risk of potentially avoidable visits by PCMH attribution type, children in the ADD cohort that attended medical homes exhibited a 3.2 to 7.6 percentage point increase in risk of a potentially avoidable visit so long as proximity to primary care providers was more convenient than EDs (p < 0.05). There were no differences in risk of an avoidable ED visit among the ADD cohort when proximity to hospitals was the most convenient.

When assessed by PCMH attribution and enrollment type, children in the ADD cohort who were enrolled in a medical home, but never attended one for their primary care services

**Table 1. Person-years summary statistics for ADD outcomes.**

| Characteristic | ADD cohort | | | | | | | |
| --- | --- | --- | --- | --- | --- | --- | --- | --- |
| | PCMH attribution based on attendance | | | PCMH attribution based on enrollment and attendance | | | | |
| | PCMH (n = 1,140) | Non-PCMH (n = 1,519) | p | Group 1 (n = 1,320) | Group 22 (n = 924) | Group 3 (n = 199) | Group 4 (n = 516) | p value |
| Age (SD) | 9.7 (1.7) | 9.8 (1.8) | 0.038 | 9.8 (1.8) | 9.6 (1.7) | 10.1 (1.7) | 9.8 (1.7) | 0.012 |
| Sex (Male) | 953 (66.2) | 999 (65.8) | 0.812 | 874 (66.2) | 620 (67.1) | 125 (62.8) | 333 (64.5) | 0.590 |
| Race | | | 0.000 | | | | | 0.000 |
| non-Hispanic White | 573 (39.8) | 721 (47.5) | | 645 (48.9) | 376 (40.7) | 76 (38.2) | 197 (38.2) | |
| Black | 820 (56.9) | 747 (49.2) | | 631 (47.8) | 524 (56.7) | 116 (58.3) | 296 (57.4) | |
| Hispanic | 47 (3.3) | 51 (3.4) | | 44 (3.3) | 24 (2.6) | 7 (3.5) | 23 (4.5) | |
| Clinical Risk Group | | | 0.547 | | | | | 0.168 |
| Healthy/non-users | 310 (21.5) | 362 (22.9) | | 304 (23.1) | 207 (22.4) | 58 (29.2) | 103 (20.0) | |
| History of significant acute disease | 50 (3.5) | 56 (3.7) | | 54 (4.1) | 29 (3.1) | 2 (1.0) | 21 (4.1) | |
| Single minor chronic disease | 594 (41.3) | 627 (41.4) | | 546 (41.5) | 394 (42.7) | 81 (40.7) | 200 (38.8) | |
| Minor chronic disease in multiple organ systems | 53 (3.7) | 46 (3.0) | | 41 (3.1) | 30 (3.3) | 5 (2.5) | 23 (4.5) | |
| Single dominant or moderate chronic disease | 319 (22.2) | 326 (21.5) | | 282 (21.4) | 188 (20.4) | 44 (22.1) | 131 (25.4) | |
| Significant chronic disease in multiple organ systems | 109 (7.6) | 97 (6.4) | | 88 (21.4) | 72 (7.8) | 9 (4.5) | 37 (7.2) | |
| Dominant chronic disease in 3+ organ systems | 1 (0.1) | 0 (0.0) | | 0 (0.0) | 1 (0.1) | 0 (0.0) | 0 (0.0) | |
| Dominant and metastatic malignancies | 1 (0.1) | 0 (0.0) | | 0 (0.0) | 0 (0.0) | 0 (0.0) | 1 (0.2) | |
| Catastrophic condition status | 2 (0.1) | 1 (0.1) | | 1 (0.1) | 2 (0.2) | 0 (0.0) | 0 (0.0) | |
| FFS plan | 105 (7.3) | 142 (9.4) | 0.043 | 139 (10.5) | 98 (10.6) | 3 (1.5) | 7 (1.4) | 0.000 |
| Primary care visits (SD) | 4.2 (2.3) | 3.7 (2.0) | 0.000 | 3.8 (2.1) | 4.2 (2.3) | 3.5 (1.9) | 4.2 (2.1) | 0.000 |
| Weekend ED visits | 403 (28.0) | 410 (27.0) | 0.545 | 351 (26.6) | 258 (27.9) | 59 (29.7) | 145 (28.1) | 0.757 |
| Dwelling location | | | 0.000 | | | | | 0.000 |
| Urban | 643 (44.9) | 504 (33.3) | | 423 (32.1) | 398 (43.4) | 81 (40.7) | 245 (47.8) | |
| Suburban | 321 (22.4) | 509 (33.6) | | 448 (34.0) | 200 (21.8) | 61 (30.7) | 121 (23.6) | |
| Rural | 467 (32.6) | 503 (33.2) | | 446 (33.9) | 320 (34.9) | 57 (28.6) | 147 (28.7) | |
| Median household income of census tract (SD) | 40,563 (13,960) | 40,595 (14,376) | 0.928 | 40,725 (14,653) | 40,077 (12,829) | 39,772 (12,454) | 41,439 (15,765) | 0.050 |
| County PCMH proportion | 21.5 (16.6) | 16.3 (15.3) | 0.000 | 15.5 (15.3) | 20.8 (16.3) | 21.8 (14.3) | 22.7 (16.8) | 0.000 |
| Mean distance to PCP (SD) | 42.7 (54.6) | 43.5 (53.9) | 0.535 | 43.7 (57.1) | 41.7 (48.8) | 42.6 (23.9) | 44.3 (63.4) | 0.565 |
| Mean distance to ED (SD) | 18.6 (55.1) | 18.5 (47.0) | 0.955 | 18.5 (49.5) | 17.9 (50.7) | 18.2 (26.7) | 19.8 (62.3) | 0.803 |
| ED visits | | | | | | | | |
| Non-emergent (avoidable) | 616 (21.0) | 739 (23.6) | 0.014 | 621 (23.0) | 383 (20.4) | 118 (28.0) | 233 (22.1) | 0.006 |
| Emergent/primary care treatable (potentially avoidable) | 514 (17.5) | 512 (16.4) | 0.235 | 442 (16.4) | 320 (17.0) | 70 (16.6) | 194 (18.4) | 0.512 |

PCMH attribution groups: (1) un-enrolled and never attended; (2) un-enrolled and attended, (3) enrolled and never attended, (4) enrolled and always attended;

PCP = primary care provider; ED = emergency department; SD = standard deviation (from mean); FFS = fee-for-service.

exhibited a 4.9 to 5.8 percentage point increase in risk of an avoidable ED visit relative to children in the reference group so long as proximity to primary care providers was more convenient (p < 0.05), whereas children who attended medical homes even though they were un-enrolled in one through their MCO showed a 7.9 to 3.1 percentage point decrease in risk of avoidable ED visits relative to children in the reference group (p < 0.05). Similar trends in risk were found among children who were enrolled and attended medical homes, but the reductions in risk were not statistically significant.

**Table 2. Person-years summary statistics for MMA outcomes.**

| Characteristic | PCMH attribution based on attendance | | | ADD cohort | | | | |
|---|---|---|---|---|---|---|---|---|
| | | | | PCMH attribution based on enrollment and attendance | | | | |
| | PCMH (n = 3,274) | Non-PCMH (n = 3,116) | p | Group 1 (n = 2,625) | Group 22 (n = 1,762) | Group 3 (n = 491) | Group 4 (n = 1,512) | p value |
| Age (SD) | 9.8 (3.5) | 10.2 (3.6) | 0.000 | 10.2 (3.6) | 9.7 (3.5) | 10.3 (3.6) | 9.8 (3.5) | 0.000 |
| Sex (Male) | 1,902 (58.1) | 1,800 (57.8) | 0.791 | 1,518 (57.8) | 1,011 (57.4) | 282 (57.4) | 891 (58.9) | 0.826 |
| Race | | | 0.098 | | | | | 0.000 |
| non-Hispanic White | 1,095 (33.5) | 1,101 (35.3) | | 953 (36.3) | 593 (33.7) | 148 (30.1) | 502 (33.2) | |
| Black | 1,952 (59.6) | 1,832 (58.8) | | 1,529 (58.3) | 1,070 (60.7) | 303 (61.7) | 882 (58.3) | |
| Hispanic | 227 (6.9) | 183 (5.9) | | 143 (5.5) | 99 (5.6) | 40 (8.2) | 128 (8.5) | |
| Clinical Risk Group | | | 0.019 | | | | | 0.036 |
| Healthy/non-users | 267 (8.2) | 291 (9.3) | | 243 (9.3) | 128 (7.3) | 48 (9.8) | 139 (9.2) | |
| History of significant acute disease | 277 (8.5) | 327 (10.5) | | 269 (10.3) | 140 (8.0) | 58 (11.8) | 137 (9.1) | |
| Single minor chronic disease | 155 (4.7) | 168 (5.4) | | 138 (5.3) | 86 (4.9) | 30 (6.1) | 69 (4.6) | |
| Minor chronic disease in multiple organ systems | 14 (0.4) | 15 (0.5) | | 12 (0.5) | 10 (0.6) | 3 (0.6) | 4 (0.3) | |
| Single dominant or moderate chronic disease | 1,970 (60.2) | 1,818 (58.3) | | 1,526 (58.1) | 1,072 (60.8) | 292 (59.5) | 898 (59.4) | |
| Significant chronic disease in multiple organ systems | 585 (17.9) | 492 (15.8) | | 433 (16.5) | 322 (18.3) | 59 (12.0) | 263 (17.4) | |
| Dominant chronic disease in 3+ organ systems | 2 (0.1) | 0 (0.0) | | 0 (0.0) | 2 (0.1) | 0 (0.0) | 0 (0.0) | |
| Dominant and metastatic malignancies | 1 (0.0) | 2 (0.1) | | 2 (0.1) | 0 (0.0) | 0 (0.0) | 1 (0.1) | |
| Catastrophic condition status | 3 (0.1) | 3 (0.1) | | 2 (0.1) | 2 (0.1) | 1 (0.2) | 1 (0.1) | |
| FFS plan | 172 (5.3) | 178 (5.7) | 0.420 | 170 (6.5) | 146 (8.3) | 8 (1.6) | 26 (1.7) | 0.000 |
| Primary care visits (SD) | 5.7 (3.5) | 4.6 (2.5) | 0.000 | 4.6 (2.4) | 5.6 (3.5) | 4.5 (2.6) | 5.8 (3.6) | 0.000 |
| Weekend ED visits | 1,061 (32.1) | 917 (29.4) | 0.010 | 762 (29.0) | 586 (33.3) | 155 (31.6) | 475 (31.4) | 0.027 |
| Dwelling location | | | 0.000 | | | | | 0.000 |
| Urban | 1,762 (53.9) | 1,330 (42.7) | | 1,065 (40.6) | 921 (52.3) | 265 (54.0) | 841 (55.6) | |
| Suburban | 711 (21.7) | 897 (28.8) | | 787 (30.0) | 379 (21.5) | 110 (22.4) | 332 (22.0) | |
| Rural | 799 (24.4) | 889 (28.5) | | 773 (30.5) | 460 (26.1) | 116 (23.6) | 339 (22.4) | |
| Median household income of census tract (SD) | 42,281 (14,801) | 40,508 (14,107) | 0.000 | 40,422 (13,908) | 41,988 (14,955) | 40,968 (15,125) | 42,636 (14,607) | 0.000 |
| County PCMH proportion | 21.1 (16.0) | 16.0 (15.0) | 0.000 | 15.3 (14.8) | 21.2 (15.9) | 19.4 (16.0) | 21.1 (16.0) | 0.000 |
| Mean distance to PCP (SD) | 36.8 (26.7) | 41.6 (31.4) | 0.000 | 41.3 (29.8) | 37.6 (28.5) | 43.1 (39.2) | 35.8 (24.1) | 0.000 |
| Mean distance to ED (SD) | 15.8 (28.3) | 17.3 (40.6) | 0.003 | 16.8 (28.7) | 16.2 (22.9) | 19.4 (78.0) | 15.2 (33.6) | 0.000 |
| ED visits | | | | | | | | |
| Non-emergent (avoidable) | 2,121 (20.8) | 2,084 (20.3) | 0.359 | 1,713 (19.8) | 1,161 (20.8) | 371 (22.9) | 960 (20.8) | 0.028 |
| Emergent/primary care treatable (potentially avoidable) | 1,730 (17.0) | 1,785 (17.4) | 0.441 | 1,508 (17.4) | 947 (17.0) | 277 (17.1) | 783 (17.0) | 0.877 |

PCMH attribution groups: (1) un-enrolled and never attended; (2) un-enrolled and attended, (3) enrolled and never attended, (4) enrolled and always attended;

PCP = primary care provider; ED = emergency department; SD = standard deviation (from mean); FFS = fee-for-service.

As shown in Table 5, there were few statistically significant associations between PCMH attribution type and risk of avoidable and potentially avoidable ED visits among the MMA cohort after including the spatial interactions. As in the a-spatial comparisons, persons who were enrolled in a medical home, but did not attend one for their primary care exhibited an increased risk of avoidable ED visits, ranging in magnitude from 2.9 to 3.7 percentage points, but only when the proximity to the hospital was closer than their primary care provider (p < 0.05). No other spatial interactions were statistically significant.

**Table 3. A-spatial comparisons of predicted probabilities in ED visit category by medical home attribution grouping type.**

| | ADD Cohort | | MMA Cohort | |
|---|---|---|---|---|
| | **Avoidable Visits** | **Potentially Avoidable Visits** | **Avoidable Visits** | **Potentially Avoidable Visits** |
| **PCMH grouping type 1** | | | | |
| Non-PCMH | 0.235 (0.008)** | 0.163 (0.007)** | 0.201 (0.004)** | 0.172 (0.004)** |
| PCMH | 0.211 (0.008)** | 0.174 (0.007)** | 0.205 (0.004)** | 0.168 (0.004)** |
| **Δ Differences in margins (type 1)** | | | | |
| PCMH versus non-PCMH | **-0.024 (0.011)*** | 0.011 (0.010) | 0.004 (0.006) | -0.004 (0.006) |
| **PCMH grouping type 2** | | | | |
| Un-enrolled + never attended (1) | 0.229 (0.009)** | 0.163 (0.008)** | 0.196 (0.005)** | 0.173 (0.005)** |
| Un-enrolled + always attended (2) | 0.205 (0.010)** | 0.169 (0.009)** | 0.206 (0.006)** | 0.167 (0.005)** |
| Enrolled + never attended (3) | 0.282 (.022)** | 0.163 (0.019)** | 0.226 (0.016)** | 0.169 (0.010)** |
| Enrolled + always attended (4) | 0.223 (0.013)** | 0.183 (0.012)** | 0.204 (0.007)** | 0.169 (0.006)** |
| **Δ Differences in margins (type 2)** | | | | |
| group 2 versus group 1 | -0.006 (0.016) | 0.020 (0.014) | 0.007 (0.008) | -0.004 (0.008) |
| group 3 versus group 1 | **0.054 (0.024)*** | 0.004 (0.020) | **0.030 (0.012)*** | -0.004 (0.011) |
| group 4 versus group 1 | -0.024 (0.013) | 0.006 (0.012) | 0.009 (0.008) | -0.006 (0.007) |

PCMH attribution groups: (1) un-enrolled and never attended; (2) un-enrolled and attended, (3) enrolled and never attended, (4) enrolled and always attended. All group comparisons in bold are statistically significant at

** $p < 0.01$,

* $p < 0.05$.

Prior to the analysis, we conducted a sensitivity analysis of ED visits by primary diagnosis codes and PCMH attribution group. For the ADD cohort, the top 10 primary diagnosis codes were identical for 90% of all visits, with the leading causes attributed to J02.9 (Acute pharyngitis, unspecified); J06.9(Acute upper respiratory infection, unspecified); and J02.0 (Streptococcal pharyngitis). Depending on the group, the top visit causes contributed to 20% to 25% of all visits. The diagnosis codes were identical for 88% of the leading visit causes for the MMA cohort, with most claims attributed to J45.901 (Unspecified asthma with (acute) exacerbation); J06.9 (Acute upper respiratory infection, unspecified); and J45.909 (Unspecified asthma, uncomplicated). Depending on group, the top visit causes accounted for 32% - 34% of all visits, respectively.

## Discussion

This study analyzed a subset of South Carolina Medicaid pediatric claims data for beneficiaries having a pre-existing diagnosis of asthma or attention-deficit/hyperactivity disorder and assessed different a-spatial and spatial interactions of visit risk based on whether or not the beneficiary attended a designated medical home. We assessed two different depictions of a beneficiary's PCMH attribution type: one based completely on attendance, and one based on enrollment and attendance records. We found that geographical location between patients, primary care providers, and hospitals does play a role in ED utilization for some pediatric groups with pre-existing chronic illnesses. We also found that these associations varied by PCMH attribution type. Many of the interactions showed that a-spatial findings were driven by instances where children had better access to medical homes versus hospitals. The intersection of these findings suggests that for some pediatric patient populations with pre-existing illnesses, a medical home effect on ED utilization rates may be contingent on where primary providers are located relative to hospitals. These findings provide some evidence that previous

**Table 4. Comparisons of predicted probabilities in ED visits for ADD outcomes by medical home attribution type after accounting for geographic proximity to providers.**

| | ADD Cohort | | | | | | | | | | | | | |
| --- | --- | --- | --- | --- | --- | --- | --- | --- | --- | --- | --- | --- | --- | --- |
| | Avoidable Visits | | | | | | | Potentially Avoidable Visits | | | | | | |
| | (Greater proximity to PCP) | | | Equal distance | (Greater proximity to ED) | | | (Greater proximity to PCP) | | | Equal distance | (Greater proximity to ED) | | |
| | 60 miles | 30 miles | 15 miles | 0 miles | 15 miles | 30 miles | 60 miles | 60 miles | 30 miles | 15 miles | 0 miles | 15 miles | 30 miles | 60 miles |
| **PCMH attribution (type 1)** | | | | | | | | | | | | | | |
| Non-PCMH | 0.255 (0.025) | 0.248 (0.017) | 0.245 (0.014) | 0.242 (0.011) | 0.239 (0.009) | 0.236 (0.008) | 0.230 (0.012) | 0.127 (0.020) | 0.139 (0.015) | 0.145 (0.012) | 0.152 (0.010) | 0.158 (0.008) | 0.165 (0.007) | 0.180 (0.013) |
| PCMH | 0.182 (0.023) | 0.192 (0.016) | 0.197 (0.013) | 0.203 (0.010) | 0.208 (0.008) | 0.213 (0.008) | 0.225 (0.014) | 0.204 (0.025) | 0.194 (0.012) | 0.189 (0.013) | 0.184 (0.010) | 0.179 (0.008) | 0.174 (0.008) | 0.165 (0.012) |
| **Δ Differences in margins** | **-0.072 (0.034)*** | **-0.056 (0.024)*** | **-0.048 (0.019)*** | **-0.039 (0.015)**\*\* | **-0.031 (0.012)**\*\* | -0.023 (0.012) | -0.005 (0.018) | **0.076 (0.032)*** | **0.055 (0.022)*** | **0.043 (0.018)*** | **0.032 (0.014)*** | 0.021 (0.011) | 0.009 (0.011) | -0.014 (0.017) |
| **PCMH attribution (type 2)** | | | | | | | | | | | | | | |
| Unenrolled + never went (1) | 0.253 (0.027) | 0.245 (0.019) | 0.241 (0.015) | 0.237 (0.011) | 0.232 (0.009) | 0.228 (0.009) | 0.221 (0.013) | 0.134 (0.023) | 0.144 (0.017) | 0.149 (0.013) | 0.154 (0.010) | 0.159 (0.008) | 0.165 (0.008) | 0.176 (0.013) |
| Unenrolled + always went (2) | 0.173 (0.026) | 0.184 (0.019) | 0.190 (0.016) | 0.195 (0.012) | 0.201 (0.010) | 0.207 (0.010) | 0.219 (0.016) | 0.193 (0.029) | 0.186 (0.020) | 0.182 (0.015) | 0.178 (0.012) | 0.174 (0.010) | 0.171 (0.010) | 0.165 (0.015) |
| Enrolled + never went (3) | 0.259 (0.067) | 0.268 (0.047) | 0.272 (0.037) | 0.277 (0.029) | 0.282 (0.023) | 0.286 (0.022) | 0.296 (0.037) | 0.082 (0.041) | 0.105 (0.034) | 0.119 (0.030) | 0.134 (0.024) | 0.151 (0.020) | 0.170 (0.020) | 0.212 (0.041) |
| Enrolled + always went (4) | 0.199 (0.045) | 0.208 (0.032) | 0.212 (0.025) | 0.216 (0.019) | 0.220 (0.014) | 0.225 (0.014) | 0.233 (0.025) | 0.227 (0.047) | 0.210 (0.030) | 0.202 (0.023) | 0.195 (0.016) | 0.187 (0.012) | 0.180 (0.012) | 0.166 (0.019) |
| **Δ Differences in margins** | | | | | | | | | | | | | | |
| group 2 versus group 1 | **-0.079 (0.038)*** | **-0.061 (0.027)*** | **-0.051 (0.021)*** | **-0.041 (0.017)*** | **-0.031 (0.014)*** | -0.021 (0.013) | -0.002 (0.021) | 0.059 (0.037) | 0.042 (0.026) | 0.033 (0.020) | 0.025 (0.016) | 0.016 (0.013) | 0.007 (0.013) | -0.011 (0.020) |
| group 3 versus group 1 | 0.006 (0.072) | 0.023 (0.050) | 0.032 (0.040) | 0.041 (0.031) | **0.049 (0.025)*** | **0.058 (0.024)*** | 0.075 (0.039) | -0.052 (0.046) | -0.039 (0.038) | -0.030 (0.033) | -0.020 (0.026) | -0.008 (0.021) | 0.005 (0.021) | 0.036 (0.044) |
| group 4 versus group 1 | -0.054 (0.053) | -0.037 (0.037) | -0.029 (0.028) | -0.020 (0.022) | -0.012 (0.017) | -0.004 (0.016) | 0.013 (0.028) | 0.093 (0.052) | **0.067 (0.034)*** | **0.054 (0.026)*** | **0.041 (0.019)*** | 0.028 (0.015) | 0.015 (0.014) | -0.010 (0.023) |

PCMH attribution groups: (1) un-enrolled and never attended; (2) un-enrolled and attended, (3) enrolled and never attended, (4) enrolled and always attended. All group comparisons in bold are statistically significant at

\*\* $p < 0.01$,

\* $p < 0.05$.

pediatric assessments of ED utilizations may be under-counting the benefit of medical home-modeled care owing to a distance decay effect patient's and providers. That medical home effectiveness gets diluted owing to geographic proximity to providers may be particularly relevant for future evaluations owing to recent findings that primary care practices that elect to transition into designated medical homes tend to locate in more socioeconomically advantaged communities [74].

**Table 5. Comparisons of predicted probabilities in ED visits for MMA outcomes by medical home attribution type after accounting for geographic proximity to providers.**

| | MMA Cohort | | | | | | | | | | | | | |
| --- | --- | --- | --- | --- | --- | --- | --- | --- | --- | --- | --- | --- | --- | --- |
| | Avoidable Visits | | | | | | | Potentially Avoidable Visits | | | | | | |
| | (Greater proximity to PCP) | | | Equal distance | (Greater proximity to ED) | | | (Greater proximity to PCP) | | | Equal distance | (Greater proximity to ED) | | |
| | 60 miles | 30 miles | 15 miles | 0 miles | 15 miles | 30 miles | 60 miles | 60 miles | 30 miles | 15 miles | 0 miles | 15 miles | 30 miles | 60 miles |
| **PCMH attribution (type 1)** | | | | | | | | | | | | | | |
| Non-PCMH | 0.191 (0.010) | 0.195 (0.007) | 0.197 (0.006) | 0.199 (0.005) | 0.201 (0.005) | 0.203 (0.004) | 0.207 (0.006) | 0.171 (0.011) | 0.173 (0.007) | 0.174 (0.006) | 0.174 (0.005) | 0.175 (0.004) | 0.176 (0.004) | 0.177 (0.006) |
| PCMH | 0.209 (0.012) | 0.209 (0.008) | 0.208 (0.007) | 0.208 (0.005) | 0.208 (0.005) | 0.208 (0.005) | 0.208 (0.007) | 0.174 (0.009) | 0.173 (0.006) | 0.172 (0.005) | 0.171 (0.005) | 0.171 (0.004) | 0.170 (0.004) | 0.168 (0.006) |
| **Δ Differences in margins** | 0.018 (0.015) | 0.014 (0.011) | 0.012 (0.009) | 0.010 (0.007) | 0.007 (0.006) | 0.005 (0.007) | 0.001 (0.009) | 0.003 (0.014) | 0.000 (0.010) | -0.001 (0.008) | -0.002 (0.007) | -0.005 (0.006) | -0.006 (0.006) | -0.009 (0.008) |
| **PCMH attribution (type 2)** | | | | | | | | | | | | | | |
| Unenrolled + never went (1) | 0.190 (0.010) | 0.192 (0.008) | 0.194 (0.006) | 0.195 (0.005) | 0.196 (0.005) | 0.198 (0.005) | 0.201 (0.006) | 0.167 (0.010) | 0.170 (0.008) | 0.172 (0.006) | 0.173 (0.005) | 0.175 (0.005) | 0.177 (0.005) | 0.180 (0.007) |
| Unenrolled + always went (2) | 0.198 (0.017) | 0.202 (0.012) | 0.204 (0.009) | 0.206 (0.007) | 0.208 (0.006) | 0.210 (0.006) | 0.214 (0.010) | 0.164 (0.014) | 0.166 (0.010) | 0.167 (0.008) | 0.168 (0.006) | 0.169 (0.006) | 0.170 (0.006) | 0.172 (0.008) |
| Enrolled + never went (3) | 0.207 (0.014) | 0.214 (0.018) | 0.218 (0.016) | 0.222 (0.014) | 0.226 (0.012) | 0.230 (0.012) | 0.238 (0.015) | 0.186 (0.032) | 0.181 (0.022) | 0.179 (0.017) | 0.177 (0.014) | 0.175 (0.011) | 0.173 (0.010) | 0.168 (0.015) |
| Enrolled + always went (4) | 0.217 (0.014) | 0.213 (0.010) | 0.212 (0.009) | 0.210 (0.007) | 0.209 (0.007) | 0.207 (0.007) | 0.204 (0.010) | 0.181 (0.010) | 0.177 (0.008) | 0.176 (0.007) | 0.174 (0.006) | 0.172 (0.006) | 0.170 (0.006) | 0.166 (0.007) |
| **Δ Differences in margins** | | | | | | | | | | | | | | |
| group 2 versus group 1 | 0.008 (0.020) | 0.009 (0.014) | 0.010 (0.011) | 0.011 (0.009) | 0.011 (0.008) | 0.012 (0.008) | 0.013 (0.012) | -0.003 (0.017) | -0.004 (0.012) | -0.005 (0.010) | -0.005 (0.008) | -0.006 (0.007) | -0.007 (0.007) | -0.008 (0.011) |
| group 3 versus group 1 | 0.017 (0.026) | 0.022 (0.020) | 0.025 (0.017) | 0.027 (0.015) | **0.029 (0.013)**\* | **0.032 (0.013)**\* | **0.037 (0.016)**\* | 0.018 (0.033) | 0.011 (0.023) | 0.007 (0.018) | 0.004 (0.015) | -0.000 (0.012) | -0.004 (0.011) | -0.012 (0.016) |
| group 4 versus group 1 | 0.027 (0.017) | 0.021 (0.013) | 0.018 (0.010) | 0.015 (0.009) | 0.012 (0.009) | 0.009 (0.008) | 0.003 (0.001) | 0.015 (0.014) | 0.008 (0.011) | 0.004 (0.009) | 0.000 (0.008) | -0.003 (0.008) | -0.007 (0.008) | -0.014 (0.010) |

PCMH attribution groups: (1) un-enrolled and never attended; (2) un-enrolled and attended, (3) enrolled and never attended, (4) enrolled and always attended. All group comparisons in bold are statistically significant at

\*\* p < 0.01,

\* p < 0.05.

However, despite these findings, the effects of geographic location on risk reductions attributable to medical home attendance were not consistently found for both cohorts. Contextually, the mixed findings raise questions as to why PCMH attendance would exhibit a benefit to some ED visit types and opposite trends for other visit types once proximity to providers was assessed. That the spatial interactions either increased the a-spatial finding that PCMH attendance lowered risk of ED utilization or revealed trends that were not previously observed suggests that the significance of geographic location is more meaningful than spurious. One explanation might stem from the differences in proximity to providers within the MMA cohort, as its medical home attendees had to travel approximately 5 to 10 miles less to access

PCMHs compared to children in the ADD cohort. Patient morbidity, provider recommendations, or time of day may have also accounted for differences in utilization trends for emergent but potentially avoidable visits. Whether these patterns are repeated in other patient cohorts or among adult ED utilization patterns is an important topic for future research.

Of the recent summaries and meta-analyses of medical home evaluations, the reviews conducted by Sinaiko *et al* (2017), Friedberg *et al* (2014), and Jackson *et al* (2013) have shown variation in overall improvements in primary care utilization, ED visits, as well as inpatient hospital admissions [41,75,76]. Factors such as practice size, patient mix, and practice ownership contribute to observed heterogeneity in study findings. Although evidence exists that geography escalates variation in healthcare access and outcomes [77–83], geographic location as a potential cause for variation in PCMH success has not been rigorously examined. This gap is significant as needs-based medical assistance programs such as the Centers for Medicare and Medicaid Services (CMS) rely on information technology platforms such as GIS to evaluate whether Medicaid beneficiaries have timely and adequate access to the providers and services that participate in its network [58,84]. Our findings that higher utilization patterns may have been driven by relative distances to providers or that non-emergent visits were consistently higher among patients that were documented as enrolled in a PCMH but never attended one for their primary care may point to specific instances where network adequacy thresholds that account for relative distances between providers might be warranted. Neither of these findings could have been uncovered without including spatial interactions in the models.

The HEDIS measures chosen for analysis were selected based on the relationship between comorbid conditions and hospital emergency department encounters that are often avoidable or treatable in primary care settings [42,43]. Its use also helped establish baseline similarities across the patient panels that made it possible to exploit differences in outcomes attributed to medical home status and/or geography. Although our study was limited in the number of quality indicators available from the encounter data, our findings are encouraging from the standpoint that additional benefits (or lack thereof) of medical home attribution can be identified for patient populations once details of health services locations are included in the model. Future work could build on this approach to assess whether the geographic location of medical homes modifies other performance measures.

The observations from this study have some noteworthy limitations. Firstly, access to health care services is a multidimensional concept and influenced by factors that have both spatial and a-spatial dimensions. Although our evaluation did allow us to control for key factors required for meaningful risk-adjustment, other clinical and patient-reported measures that we were unable to obtain may further delineate the benefits attributed to medical home access. For example, we could not control for factors such as wait times, historical service use, time or day, or patient satisfaction with ED care versus primary care in our analysis, all of which may have helped to further contextualize differences in use. Similarly, although many state Medicaid agencies are experimenting with some version of PCMH transformation, the varied nature of state Medicaid characteristics make national geographic comparisons difficult to construct and we cannot say whether these trends over or underestimate its effect. Additional state-based tests would be beneficial for context validity as well as for defining appropriate access thresholds for other performance measures. With respect to our spatial approach, the primary limitation of assessing relative differences to providers is that we did not focus on exact distances to care. For example, in our model someone could have been assigned into the 0–15 mile grouping if they lived 20 miles from the ED or 200 miles from the ED, so long as the difference in proximity to their primary care provider remained within 15 miles. Similarly, our analysis may have been biased owing to the lack of information on PCMH enrollment and attendance in the claims data, although we were able to circumvent some of these issues using

the enrollment and attendance matrix. Similarly, we could not confirm whether all beneficiaries initiated their care from their place of residence or how they traveled to obtain care (e.g., car, bus, walk), but we could confirm that all claims represented physical encounters at primary care clinics and hospitals. Another limitation is that our analysis may have muted disparities that might have emerged had we included privately insured populations. However, we did find that differences in outcomes among some of the most socially vulnerable populations in the state could be demarcated once the models accounted for relative distances populations must travel to obtain health care services.

## Conclusion

In the age of patient-centered medicine, GIS technology is becoming even more important for optimizing case management (e.g., list of available transportation resources) as well as enhancing patient experience with care (e.g., providing web-based service locations of nearby providers). Although geographic variation is an accepted phenomenon in health care services research, the findings in this study illustrate that health care performance measures can be enhanced by knowing how far patients must travel in order to obtain services. Although medical home attendance may not be consistently associated with lower ED visits for all patient groups, our findings suggest that the lack of spatial information may be one factor why the anticipated effects of some PCMH innovations is often muted.

## Supporting information

**S1 Table. Adjusted GEE coefficients for ED visits where PCMH attribution was based on attendance data.** Covariates were included in the models if bivariate comparisons had p-values at or below 0.10.
(DOCX)

**S2 Table. Adjusted GEE coefficients for ED visits where PCMH attribution was based on attendance and enrollment data.** Covariates were included in the models if bivariate comparisons had p-values at or below 0.10.
(DOCX)

## Author Contributions

**Conceptualization:** Nathaniel Bell, Ana Lòpez-De Fede.

**Formal analysis:** Nathaniel Bell.

**Funding acquisition:** Ana Lòpez-De Fede, Bo Cai, John Brooks.

**Methodology:** Nathaniel Bell, Ana Lòpez-De Fede, Bo Cai, John Brooks.

**Project administration:** Ana Lòpez-De Fede.

**Writing – original draft:** Nathaniel Bell, Ana Lòpez-De Fede, Bo Cai, John Brooks.

**Writing – review & editing:** Nathaniel Bell, Ana Lòpez-De Fede, Bo Cai, John Brooks.

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
