## [Decision Letter · Decision Letter 0]

25 Apr 2022

PONE-D-22-04552Geographic proximity to primary care providers as a risk-assessment criterion for quality performance measuresPLOS ONE

Dear Dr. Bell,

Thank you for submitting your manuscript to PLOS ONE. After careful consideration, we feel that it has merit but does not fully meet PLOS ONE’s publication criteria as it currently stands. Therefore, we invite you to submit a revised version of the manuscript that addresses the points raised during the review process.

We look forward to receiving your revised manuscript.

Kind regards,

Jingjing Qian

Academic Editor

PLOS ONE

Journal Requirements:

1. You may seek permission from the original copyright holder of Figure(s) [#] to publish the content specifically under the CC BY 4.0 license.  

Reviewers' comments:

Reviewer's Responses to Questions

**Comments to the Author**

1. Is the manuscript technically sound, and do the data support the conclusions?

Reviewer #1: Yes

Reviewer #2: Partly

2. Has the statistical analysis been performed appropriately and rigorously? 

Reviewer #1: I Don't Know

Reviewer #2: No

3. Have the authors made all data underlying the findings in their manuscript fully available?

Reviewer #1: No

Reviewer #2: Yes

4. Is the manuscript presented in an intelligible fashion and written in standard English?

Reviewer #1: Yes

Reviewer #2: Yes

5. Review Comments to the Author

Reviewer #1: Summary and Overall Impression

Thank you for the opportunity to review this important work.

This work focused attention to an evolving area of health services research that is much needed to non-medical determinants of health. The authors appear to take a very thorough approach in the development of their models to explore the possible linkage between PMCH designation and geographic distance to primary care clinic ED. The use of proxy PCMH attribution status seems relevant and plausible in the absence of an alternative for its proposed application, however, the interpretation of the results given the age groups (children) and uncertainty around these group designations (ie: enrolled vs assigned to PCMH) makes one given some consideration to the generalizability of the findings. The discussion appropriately highlights some key aspects of the work that have relevance to the existing literature, some inherent weaknesses mainly in the short-comings of using administrative data and future work.

As a reader/reviewer, my impression during an intense second read, it became apparent that some of the discussion contained important contextual (background) information that likely could be used to strengthen the introduction - and provide additional points for the reader to better understand the 'how' and 'why' for the analysis was undertaken. I believe this will contextualize the results in a more purposeful fashion. For example: providing additional background the states of S. Carolina, it's population and those on medicaid along with existing rural - urban disparities in access to a family physician. (i.e. what % of the rural population are served by 'X' number of doctors vs those in urban areas). And, for the demographics chosen, are there options also for care to be received by community Pediatricians outside the PCMH umbrella. With some additional focused information on the geographic and demographic variable, perhaps framed in a hypothesis, will greatly support the outcomes found in this work.

Minor

Lines 13-22 This second paragraph contains alot of important information some of which does not

relate directly to the study. Consider a much shorter segue to third paragraph to further focus the message.

Table 3 - contains some valuable information; suggest CRG values assigned be displayed vs descriptive breakdown

for ease of viewing and interpretation.

Miscellaneous

Table 1 - Distance to PCP & Distance to ED - is the average distance?. Further clarification might be helpful.

The sentence in Lines 9-11 ending with 'healthy and prosperous society' seems to be a key statement setting up the study; with some minor editing could be strengthened to highlight to the reader the interplay of (inequity) access to primary care, SDoH and improved patient health outcome and overall population health vs. policies for healthy and prosperous society' any referenced work from Sir Michael Marmot or Barbara Starfield might be valuable to consider.

Lines 18-20 re: Ontario Family Health Team indicates 'specialty care teams', is this accurate, versus community primary care providers working along side allied health (social work, psychologist) and nursing? Specialty care suggests surgery or medical specialist providers.

P-values in results descriptions ex: Pg 10 Lines 12-14 p-values are missing.

Pg10 Line 14 medical ? home.

Reviewer #2: In this three-year retrospective cohort study, authors evaluated the number of pediatric ED visits based on PCMH status and geographic location to their PCP or hospital. Authors concluded an interesting finding that the PCMH model may not lower ED visits but geographic proximity to care may need to be considered. Overall, the manuscript is well-written and addresses a significant issue of health disparities. However, the results do not seem to support the authors’ conclusions. For example, Table 4 provides detailed findings that, overall, do not seem to have significance between PCCMH status nor geographic location; there are minimal differences between PCMH status group comparisons including group 4 that are the most established. While the data are interesting, authors need to emphasis the major significant findings in the abstract, results, and tables so readers will clearly understand the value of this work and the gaps in literature it addresses.

GENERAL

- Change “ED admissions” to “ED visits” throughout the paper. Admission is admitted to the hospital; ED visits are patients coming to the ED but may or may not be admitted to the hospital.

- Clarification is needed of the phrase “geographic proximity to primary care providers vs hospitals”. Does this mean that the closer individuals were to the type of healthcare entity, the more likely to receive care from that place, e.g., closer to PCP then receive care from them, closer to ED, then receive care from them? If so, this seems intuitive.

ABSTRACT

- Add participant age to Methods

- Results need numerical detail, e.g., population n, % or n of findings, etc. and the corresponding statistical significance as applicable.

INTRODUCTION – overall well-written and outlines the background to the study.

- It is a strength of the study that data were collected prior to COVID since, during the pandemic, it may have been have skewed the data toward ED visits. Authors may want to omit COVID-19 from the intro (pg. 3, line 3) so readers are not confused and highlight this strength in the discussion.

- Pg 2 Line 10. Change “racism” to “race/ethnicity”

- Add the gap authors are filling from this study and the study aim/objective

METHODS – overall detailed and use validated measures for outcomes (NRU ED visits algorithm)

RESULTS

- add numerical values for major findings and their significance

- Typo on Pg 10, line 3: delete among

TABLES/FIGURES

- ensure all abbreviations are defined, e.g., MMA, ADD, HEDIS, etc. and add legends to help the reader understand the meaning/rationale of MMA, ADD, attribution groups, etc.

- Table 4: why is statistical significance considered p<0.10?

- Table 4: There are several bolded items that are not statistically significant

- Table 4: p-values are a probabilities and should not be negative

6. PLOS authors have the option to publish the peer review history of their article (what does this mean?). If published, this will include your full peer review and any attached files.

Reviewer #1: **Yes: **Terrence McDonald, MD, MSc, CCFP (SEM), FCFP, Dip. Sport Med

Reviewer #2: No

---

## [Author Response · Author response to Decision Letter 0]

6 Jul 2022

I have uploaded (as a word file) an itemized list of our response to the reviewer comments. I am pasting this information here if more convenient. Because HTML will drop formatting, the copy/past should indent our response (plus add a bullet). 

June 20, 2022

To the editors of PLOS One,

Thank you for the opportunity to have our manuscript peer-reviewed. We have revised our manuscript based on the thoughtful comments and suggestions from the reviewers and wish to extend our appreciation for their time and effort. All of their comments and suggestions have been addressed in our revised submission. 

To summarize: This study evaluated the association between patient-centered medical home (PCMH) enrollment and avoidable ED visits among pediatric Medicaid beneficiaries in South Carolina over a 3-year period prior to the COVID-19 pandemic. It addresses a limitation of ongoing medical home evaluations that suggests they are not effectively reducing avoidable ED visits among children with chronic illness. Our regression models accounted for geographic relationships between patient and primary care provider locations, which is a novel methodological contribution for evaluating these events. 

Key findings: Our study provides evidence that medical homes are helping to lower avoidable ED visits among children with pre-existing illnesses once geographical linkages between patients, providers, and hospitals are included in the models. They are important to state Medicaid agencies, payers, and clinical researchers as it points to the value of continuing to invest in medical home-modeled care as it contextualizes the benefits of PCMHs that may be going unmeasured. Our findings are also timely in light of the emergent research on social risk factor adjustment in pay-for-performance models.

We are pleased that both reviewers felt that this was important work. Briefly, the primary concern of reviewer #1 was that some key paragraphs in the discussion section be moved to the introduction to better frame the importance of this work. This resulted in a near complete re-write of the introduction, but one that we believe addresses their comments. Reviewer #1 also identified some minor methodological and grammatical errors, all of which have been addressed. Reviewer #2’s suggestions were also very pragmatic, with emphasis on clarification for why p values < 0.10 were included as well as some recommendations for reworking the abstract. These and their other minor comments have all been addressed in this revised submission. 

The following pages (attached separately) provide an itemized list of the reviewer comments (italicized) and how we amended the manuscript (bulleted) and where the changes can be found in the cleaned copy. On behalf of my co-authors, we thank you for continued consideration of this work. All authors listed in this submission fulfill the journal’s requirements for publication and gave final approval of the manuscript submitted for continued review. 

Kind regards,

Nathaniel Bell PhD

RESPONSE TO REVIEWER COMMENTS FOLLOWS

Reviewer #1: Summary and Overall Impression

Thank you for the opportunity to review this important work.

This work focused attention to an evolving area of health services research that is much needed to non-medical determinants of health. The authors appear to take a very thorough approach in the development of their models to explore the possible linkage between PMCH designation and geographic distance to primary care clinic ED. The use of proxy PCMH attribution status seems relevant and plausible in the absence of an alternative for its proposed application, however, the interpretation of the results given the age groups (children) and uncertainty around these group designations (ie: enrolled vs assigned to PCMH) makes one given some consideration to the generalizability of the findings. The discussion appropriately highlights some key aspects of the work that have relevance to the existing literature, some inherent weaknesses mainly in the short-comings of using administrative data and future work.

As a reader/reviewer, my impression during an intense second read, it became apparent that some of the discussion contained important contextual (background) information that likely could be used to strengthen the introduction - and provide additional points for the reader to better understand the 'how' and 'why' for the analysis was undertaken. I believe this will contextualize the results in a more purposeful fashion. For example: providing additional background the states of S. Carolina, it's population and those on medicaid along with existing rural - urban disparities in access to a family physician. (i.e. what % of the rural population are served by 'X' number of doctors vs those in urban areas). And, for the demographics chosen, are there options also for care to be received by community Pediatricians outside the PCMH umbrella. With some additional focused information on the geographic and demographic variable, perhaps framed in a hypothesis, will greatly support the outcomes found in this work.

• Thank you for these thoughtful comments and critiques. We have addressed each of your comments (above) in the revised manuscript.

• The entire introduction section has been heavily edited and re-written to better emphasize the ‘why’ and the ‘how’ and situate the reader within the context of what has been done, what are the major gaps, and why this work is important. These changes can be found on pages 2 through 5, which spans the entire introductory section. 

• The methods sub-section for “PCMH attribution” on page 6 was edited to concisely explain why we chose these patient attribution groups. We do agree that streamlining some of these comparisons was necessary. As such, in the revised submission we limit our comparisons to those in which PCMH is based solely on attendance (attended vs. did not attend) and limit our original enrollment/attendance comparisons to be in reference to those who were unenrolled and never attended. 

• The introduction section is now significantly longer, moving from broad to general while and emphasizes the importance of this work within the context of risk adjustment and in relation to medical home evaluations. We believe that our reframing of the introduction section based on your thoughtful comments and suggestions gets at the heart of the major comments. 

• We have added additional information on SC Medicaid statistics as well as the state’s enrollment policies for medical homes. The major changes can be found on lines 15 – 23 on page 6 as well as lines 1 – 3 on page 7 in the methods sub-section on “PCMH attribution”. Note, we do not have every statistic available that reviewer #1 is mentioning, but we have added a number of contextual points that they are mentioning to better situate the reader within the context of this work. 

Minor

Lines 13-22 This second paragraph contains a lot of important information some of which does not relate directly to the study. Consider a much shorter segue to third paragraph to further focus the message.

• Thank you for this suggestion. We have revised the entire introduction section to summarize the key areas that readers need to be aware of that are relevant to this work (1) avoidable ED utilizations, (2) primary care models designed to minimize their impact, (3) gaps in evidence and methodological approaches (e.g., GIS) that may help address the gaps, and (4) our aims/objectives for this research and why it is important and relevant. 

Table 3 - contains some valuable information; suggest CRG values assigned be displayed vs descriptive breakdown for ease of viewing and interpretation.

• We agree – thank you for this feedback. We have amended the descriptive statistics tables as well as the logistic regression tables to show how the groups were similar and different. 

• Because of the need to have a larger number of tables than average, we have provided some information (e.g., regression coefficients, standard errors) in an online appendix. 

Miscellaneous

Table 1 - Distance to PCP & Distance to ED - is the average distance? Further clarification might be helpful.

• Yes, these are average distance values to providers. We have amended lines 7-9 on page 9 to articulate this to the reader in a revised sentence that reads: “Average travel distance estimates (in miles) for all visits were then constructed for all unique pairings between the beneficiary’s residential address recorded in the claims database to the location where their primary care and ED visit occurred.”

• We have also rewritten this paragraph (page 9, lines 1-14) to more clearly explain how relative differences in proximity were assessed. 

The sentence in Lines 9-11 ending with 'healthy and prosperous society' seems to be a key statement setting up the study; with some minor editing could be strengthened to highlight to the reader the interplay of (inequity) access to primary care, SDoH and improved patient health outcome and overall population health vs. policies for healthy and prosperous society' any referenced work from Sir Michael Marmot or Barbara Starfield might be valuable to consider.

• Two very important scholars, indeed! Richard Wilkerson’s and Nancy Krieger’s work as well as Robert Evans and Clyde Hertzman’s work (for a Canadian perspective) are also key and I have found memories of meeting with them while attending graduate school at SFU and UBC. We have expanded the number of relevant references in the revised manuscript, but tried to keep the focus on studies that are specific to ED utilization.

Lines 18-20 re: Ontario Family Health Team indicates 'specialty care teams', is this accurate, versus community primary care providers working alongside allied health (social work, psychologist) and nursing? Specialty care suggests surgery or medical specialist providers.

• Yes and no. PCMH are unique to US, but other countries/systems are transitioning (or have transitioned) to team-based care, such as Canada’s Family Health Care teams. These edits can be found on page 2, lines 23-24. 

P-values in results descriptions ex: Pg 10 Lines 12-14 p-values are missing.

• Our apology for this oversight. The results section in the revised submission contains p values (when significant) and points to all tables for specific p-values or threshold p values (e.g., < p 0.01, < p < 0.05). 

Pg10 Line 14 medical ? home.

• Correct. Our apology for this typo. The sentence now includes “medical home”

Reviewer #2: Summary and Overall Impression

In this three-year retrospective cohort study, authors evaluated the number of pediatric ED visits based on PCMH status and geographic location to their PCP or hospital. Authors concluded an interesting finding that the PCMH model may not lower ED visits but geographic proximity to care may need to be considered. Overall, the manuscript is well-written and addresses a significant issue of health disparities. However, the results do not seem to support the authors’ conclusions. For example, Table 4 provides detailed findings that, overall, do not seem to have significance between PCCMH status nor geographic location; there are minimal differences between PCMH status group comparisons including group 4 that are the most established. While the data are interesting, authors need to emphasis the major significant findings in the abstract, results, and tables so readers will clearly understand the value of this work and the gaps in literature it addresses.

• Thank you for this feedback and observations. We have addressed the mismatch between the abstract + results section in the revised submission. 

• The abstract results section was revised to better summarize the main findings.

• Please note, in meeting recommendations from Reviewer #1, we amended our PCMH grouping to add two different scenarios: one based on attendance (i.e., PCMH vs non-PCMH) and one based on enrollment + attendance (i.e., identical to our original submission), so some significant changes occurred in the revised manuscript. 

GENERAL

Change “ED admissions” to “ED visits” throughout the paper. Admission is admitted to the hospital; ED visits are patients coming to the ED but may or may not be admitted to the hospital.

• We agree. All references to “admissions” have been changed to “visits” throughout. Thank you for this observation.

Clarification is needed of the phrase “geographic proximity to primary care providers vs hospitals”. Does this mean that the closer individuals were to the type of healthcare entity, the more likely to receive care from that place, e.g., closer to PCP then receive care from them, closer to ED, then receive care from them? If so, this seems intuitive.

• Yes, your interpretation is correct. We have amended the “Geographic assessment” sub-section of the Methods section (lines 1 – 14, page 9) to confirm that we are looking at relative distances to one provider over another, which adds a layer of depth to solely looking at distance to EDs or PCPs. This has its limitations, which we have added in the amended limitations section on pages 21 (lines 21- 24, and continued onto page 22) in the last paragraph of discussion section. 

ABSTRACT

Add participant age to Methods

• The age groups of the pediatric cohort have been added to the Methods section of the abstract. These age groups are also provided in the methods section (lines 5-9, page 6).

Results need numerical detail, e.g., population n, % or n of findings, etc. and the corresponding statistical significance as applicable.

• Thank you for this observation. The results section of the abstract has been revised so that the key %’s/values are shown numerically. 

INTRODUCTION 

overall well-written and outlines the background to the study. 

• Thank you. This has been an enjoyable and rewarding project to be a part of and we’re excited to share our findings. 

It is a strength of the study that data were collected prior to COVID since, during the pandemic, it may have been have skewed the data toward ED visits. Authors may want to omit COVID-19 from the intro (pg. 3, line 3) so readers are not confused and highlight this strength in the discussion.

• Great point – we have added in a sentence in the introduction section (line 16 page 2) and again in the opening line of the Methods section (line 15 page 5) to emphasize that these trends/limitations that we’re comparing and trying to address are all pre-COVID. 

Pg 2 Line 10. Change “racism” to “race/ethnicity”

• Thank you – change was made through deletion of the original paragraph during the re-write. 

Add the gap authors are filling from this study and the study aim/objective

• Thank you – the last two paragraphs of the introduction (page 4 and 5) section summarize the gap(s) that this study is seeking to address, with the preceding paragraphs providing the rationale for our approach. 

METHODS 

overall detailed and use validated measures for outcomes (NYU ED visits algorithm)

• Thank you for this feedback. 

RESULTS

add numerical values for major findings and their significance

• Thank you – percentage point differences that were significant are listed as well as their corresponding p values throughout the results section.

Typo on Pg 10, line 3: delete among

• Thank you – this grammatical error was removed during the re-write. 

TABLES/FIGURES

ensure all abbreviations are defined, e.g., MMA, ADD, HEDIS, etc. and add legends to help the reader understand the meaning/rationale of MMA, ADD, attribution groups, etc.

• Thank you. These have been added throughout when possible at the bottom of the tables. 

• We have also included a list of abbreviations at the end of the manuscript. 

Table 4: why is statistical significance considered p<0.10?

• This is a longer discussion, but the short end is that many statisticians are trying to get health services researchers/clinicians away from thinking primarily in p values and looking more specifically at the point estimates and the upper/lower bounds of these estimates. For example, take for example a point estimate/risk ratio of 1.45, with an upper bound of 1.94 and a lower bound of 0.98. Obviously, the p value for this association is going to be greater than 0.05 because it is crossing 1.00, but it (in our view) is wrong to say that there is no effect – it’s just that not all data points are compatible with the data. It is more informative to say that “Our results suggest a 45% increase in risk of an avoidable ED visit in patients who do not attend a PCMH. Nonetheless, a risk difference ranging from a 2% decrease, a small negative association, to a 94% increase, a substantial positive association, is also reasonably compatible with our data”. Interpreting the point estimate while acknowledging its uncertainty is a better way to keep us from making false declarations of “no difference” and from making overconfident claims. 

• However, to your point, this is still not the predominant view and we have revised our submission to only include p values of 0.05 or lower. 

Table 4: There are several bolded items that are not statistically significant

• Thank you - see response above. All values in bold now only correspond to associations with p values < 0.05

Table 4: p-values are a probabilities and should not be negative

• The (-) negative values in table 4 (revised table 4 and table 5) do not refer to p-values. They refer to difference in predicted probabilities between the groups. So negative values suggest a “PCMH effect” of lower ED rate, whereas positive values represent the opposite (e.g., PCMH outcomes are worse than non-PCMH outcomes). The difference in margins were tested and p-values are reported for differences that have p values less than 0.05.

---

## [Decision Letter · Decision Letter 1]

2 Aug 2022

PONE-D-22-04552R1Geographic proximity to primary care providers as a risk-assessment criterion for quality performance measuresPLOS ONE

Dear Dr. Bell,

Thank you for submitting your manuscript to PLOS ONE. After careful consideration, we feel that it has merit but does not fully meet PLOS ONE’s publication criteria as it currently stands. Therefore, we invite you to submit a revised version of the manuscript that addresses the points raised during the review process.

We look forward to receiving your revised manuscript.

Kind regards,

Jingjing Qian

Academic Editor

PLOS ONE

Journal Requirements:

Additional Editor Comments:

Thank for addressing the majority of reviewers' comments. Please make further minor edits as suggested by reviewer #2.

Reviewers' comments:

Reviewer's Responses to Questions

**Comments to the Author**

1. If the authors have adequately addressed your comments raised in a previous round of review and you feel that this manuscript is now acceptable for publication, you may indicate that here to bypass the “Comments to the Author” section, enter your conflict of interest statement in the “Confidential to Editor” section, and submit your "Accept" recommendation.

Reviewer #1: All comments have been addressed

Reviewer #2: (No Response)

2. Is the manuscript technically sound, and do the data support the conclusions?

Reviewer #1: Yes

Reviewer #2: Yes

3. Has the statistical analysis been performed appropriately and rigorously? 

Reviewer #1: Yes

Reviewer #2: I Don't Know

4. Have the authors made all data underlying the findings in their manuscript fully available?

Reviewer #1: Yes

Reviewer #2: Yes

5. Is the manuscript presented in an intelligible fashion and written in standard English?

Reviewer #1: Yes

Reviewer #2: Yes

6. Review Comments to the Author

Reviewer #1: Overall well written and is of great value adding to the understanding of the impact of the PCMH

by assessing geographic proximity and patient outcomes.

The introduction is much more focused and reads well.

Methods again appear appropriately selected and applied.

Results appears in order well-presented.

Discussion reads well and addresses limitations and outlines future work.

Minor

Pg 11 Line 21 ..provider ? 'over' another.

Pg 14 Line 15 ..care that ? 'is' considered...

Reviewer #2: 1. Rationale for the study improved with the revision but participant rationale is needed. Specifically, it is not clear why only individuals with ADD and asthma were included since the study evaluates health disparities by geographic proximity and avoidable ED visits. Examples/suggestions:

a. Abstract: In the background, add the need for studying ADD, Asthma; in the objective, add “potentially avoidable ED visits for children with ADD and asthma”. The conclusion (and manuscript) should state ADD and asthma since not all preexisting illnesses/chronic diseases were studied.

b. Introduction: (pg. 2 line 7), provide rationale for ADD and asthma (and perhaps omit homeless and diabetes) as there are many other chronic diseases that utilize EDs

2. Abstract: The conclusion is difficult to understand. Suggest 2-3 sentences stating what authors found and conclude regarding the study aim and results as well as its impact. The summary in authors’ cover letter provides much clearer conclusions.

7. PLOS authors have the option to publish the peer review history of their article (what does this mean?). If published, this will include your full peer review and any attached files.

Reviewer #1: **Yes: **Terrence McDonald, MD, MSC, FCFP

Reviewer #2: No

---

## [Author Response · Author response to Decision Letter 1]

9 Aug 2022

Reviewer #1: Summary and Overall Impression

Reviewer #1: Overall well written and is of great value adding to the understanding of the impact of the PCMH by assessing geographic proximity and patient outcomes.

The introduction is much more focused and reads well.

Methods again appear appropriately selected and applied.

Results appears in order well-presented.

Discussion reads well and addresses limitations and outlines future work.

Minor

Pg 11 Line 21 ..provider ? ‘over’ another.

• Thank you for this observation. This sentence was actually from the original submission, not from revision 1. No changes were made as the sentence had been removed in our resubmission. 

Pg 14 Line 15 ..care that ? ‘is’ considered…

• Thank you for this observation. The sentence (Page 8, Line 14 in methods section) has been amended and now reads: “…particularly care that is considered to be of higher quality” 

Reviewer #2: Summary and Overall Impression

Reviewer #2: 1. Rationale for the study improved with the revision but participant rationale is needed. Specifically, it is not clear why only individuals with ADD and asthma were included since the study evaluates health disparities by geographic proximity and avoidable ED visits. Examples/suggestions:

• Thank you for mentioning this. We have amended the Methods section (page 8 of Marked Copy, Paragraph 2 and Paragraph 3) to include an additional information on the HEDIS measures and what they are and why they are important. 

• Our original submission specified that these two measures (ADD, MMA) were chosen because they represent two disease conditions that are often treated in hospital EDs for care that could have been obtained by primary care providers. We have added an additional sentence (Page 8, lines 19 – 21) that re-emphasize this point.

a. Abstract: In the background, add the need for studying ADD, Asthma; in the objective, add “potentially avoidable ED visits for children with ADD and asthma”. The conclusion (and manuscript) should state ADD and asthma since not all preexisting illnesses/chronic diseases were studied.

• We have amended the Objective sentence in the abstract. It now reads “To examine the association between geographic proximity to primary care providers versus hospitals and risk of avoidable and potentially avoidable ED visits among children with pre-existing diagnosis of attention-deficit/hyperactivity disorder or asthma. “ 

b. Introduction: (pg. 2 line 7), provide rationale for ADD and asthma (and perhaps omit homeless and diabetes) as there are many other chronic diseases that utilize EDs

• Thank you – we have left the opening paragraph alone (as it is a framing paragraph), but have modified the 5th paragraph of the introduction section (page 5, lines 6 – 9) so that the contextual information about why our study on risk adjustment included these two metrics is clear. 

• Additional information was also provided in the methods section (page 8).

2. Abstract: The conclusion is difficult to understand. Suggest 2-3 sentences stating what authors found and conclude regarding the study aim and results as well as its impact. The summary in authors’ cover letter provides much clearer conclusions.

• Thank you for this observation. We have revised the conclusion section of the abstract. It now reads “In several health care performance evaluations, patient-centered medical homes have not been found to reduce differences in hospital utilization for conditions that are treatable in primary care settings among children with chronic illnesses. Analytical approaches that also consider geographic proximity to health care services can identify performance benefits of medical homes. Expanding risk-adjustment models to also include geographic data would benefit ongoing quality improvement initiatives.”

---

## [Editor Report · Decision Letter 2]

16 Aug 2022

Geographic proximity to primary care providers as a risk-assessment criterion for quality performance measures

PONE-D-22-04552R2

Dear Dr. Bell,

We’re pleased to inform you that your manuscript has been judged scientifically suitable for publication and will be formally accepted for publication once it meets all outstanding technical requirements.

Kind regards,

Jingjing Qian

Academic Editor

PLOS ONE

Additional Editor Comments (optional):

Thanks for the additional revision. All comments have been addressed.
---

## [Editor Report · Acceptance letter]

26 Aug 2022

PONE-D-22-04552R2 

Geographic proximity to primary care providers as a risk-assessment criterion for quality performance measures 

Dear Dr. Bell:

I'm pleased to inform you that your manuscript has been deemed suitable for publication in PLOS ONE. Congratulations! Your manuscript is now with our production department. 

Kind regards, 

on behalf of

Dr. Jingjing Qian 

Academic Editor

PLOS ONE